# Adaptive Evolution and Functional Differentiation of Testis-Expressed Genes in Theria

**DOI:** 10.3390/ani14162316

**Published:** 2024-08-09

**Authors:** Yukako Katsura, Shuji Shigenobu, Yoko Satta

**Affiliations:** 1Center for the Evolutionary Origins of Human Behavior, Kyoto University, Inuyama 484-8506, Japan; 2NIBB Core Research Facilities, National Institute for Basic Biology, Okazaki 444-0867, Japan; shige@nibb.ac.jp; 3Research Center for Integrative Evolutionary Science, SOKENDAI (The Graduate University for Advanced Studies), Hayama 240-0193, Japan; satta@soken.ac.jp

**Keywords:** testis transcriptome, marsupials, adaptive evolution, Theria, *PRDM1*, *ARHGAP28*

## Abstract

**Simple Summary:**

A transcriptome landscape of therian mammals is known, but it remains unclear how transcriptomic patterns have evolved in a marsupial- or eutherian-specific way. It is important to understand marsupial- or eutherian-specific transcriptomic patterns since their fitness and sex differentiation are different. This study examines therian testis transcriptomes to elucidate marsupial and eutherian uniqueness in male differentiation. Using the massive transcriptomic data, we show the evolutionary tempo and mode of testis-expressed genes in Theria and identify candidate genes involved in the specificity of marsupial or eutherian testes.

**Abstract:**

Gene expression patterns differ in different tissues, and the expression pattern of genes in the mammalian testis is known to be extremely variable in different species. To clarify how the testis transcriptomic pattern has evolved in particular species, we examined the evolution of the adult testis transcriptome in Theria using 10 species: two marsupials (opossum and Tasmanian devil), six eutherian (placental) mammals (human, chimpanzee, bonobo, gorilla, rhesus macaque, and mouse), and two outgroup species (platypus and chicken). We show that 22 testis-expressed genes are marsupial-specific, suggesting their acquisition in the stem lineage of marsupials after the divergence from eutherians. Despite the time length of the eutherian stem lineage being similar to that of the marsupial lineage, acquisition of testis-expressed genes was not found in the stem lineage of eutherians; rather, their expression patterns differed by species, suggesting rapid gene evolution in the eutherian ancestors. Fifteen testis-expressed genes are therian-specific, and for three of these genes, the evolutionary tempo is markedly faster in eutherians than in marsupials. Our phylogenetic analysis of *Rho GTPase-activating protein 28* (*ARHGAP28*) suggests the adaptive evolution of this gene in the eutherians, probably together with the expression pattern differentiation.

## 1. Introduction

Theria is a mammalian subclass composed of Eutheria (eutherian or placental mammals) and Metatheria (marsupials or pouched mammals), and the molecular clock suggests that these two infraclasses diverged 148 million years ago [1]. Eutheria have spread around the world, but marsupials have lived only in limited places such as the Australian and American continents, suggesting that the two groups have adapted to particular ecological environments. Differences in fitness to these environments should result from the divergence of genome and transcriptome between eutherians and marsupials. The transcriptome of reproductive tissues is directly related to the efficiency of producing offspring, and it is thus important to understand transcriptomic differences in reproductive tissues as the evolution of the directly influential system in fitness.

The therian Y chromosome is a testis determiner and possesses a sex determination gene (*SRY*). In marsupials, the molecular system of sex determination and differentiation is not as well understood as that in eutherians. We have shown that the marsupial *SRY* gene is functionally similar to eutherian *SRY* and upregulates SRY-box9 (*SOX9*) synergistically with steroidogenic factor 1 (*SF1*) [2]. This means that the system of sex determination is common in marsupials and eutherians, but the differentiation of sexual characteristics is distinct in the two groups. For example, in marsupials, gonadal differentiation occurs after birth, but in eutherians, the gonads differentiate in the embryo before birth [3]. Moreover, marsupial mammary glands and scrotum, which are sexually dimorphic structures, have usually started to differentiate before ovary and testis differentiation, independent of endogenous estrogen or testosterone [3,4,5,6,7]. Graves, Renfree, and their colleagues suggest that the marsupial Y chromosome does not control all aspects of sex differentiation, as it does in eutherians through the initiation of androgen pathways that control most sexual dimorphisms [4,6,8].

RNA sequencing (RNA-seq) using next-generation sequencers is useful for understanding the entire profile of gene expression in different tissues. Marsupial transcriptomes have been investigated by several groups [9,10,11]. Brawand et al. (2011) investigated transcriptomes of six tissues in 10 species including opossum (*Monodelphis domestica*) and showed rapid evolution of the transcriptome in testis compared to other tissues [9]. Murat et al. (2023) published transcriptomes of testes using single-cell RNA-seq and showed temporal expression changes of genes in spermatogenesis across 11 species including opossum [12]. These previous studies revealed a transcriptome landscape of mammals, but it remains unclear how transcriptomic patterns have evolved in a marsupial-specific way. In particular, we do not know which transcriptomic characteristics were acquired or lost in the stem lineage of marsupials.

In this study, we examine therian testis transcriptomes to elucidate marsupial and eutherian uniqueness in adult male differentiation. Using the massive transcriptomic data, it is now possible to understand what evolutionary events happened during therian evolution. Thus, we show the evolutionary tempo and mode of testis-expressed genes in Theria and identify candidate genes involved in the specificity of marsupial or eutherian testes.

## 2. Materials and Methods

### 2.1. Sample and Library Preparation

We used two RNA samples of Tasmanian devil testis from a single individual; these samples were kind gifts from Dr. Jenny Graves. RNA was extracted with a GenElute™ Mammalian Total RNA Miniprep Kit (Sigma, Castle Hill, NSW, Australia). cDNA libraries were generated from purified RNA (1 μg of each sample) using a TruSeq RNA Sample Preparation Kit v2 (Illumina, San Diego, CA, USA).

### 2.2. RNA Sequencing and Assembling

The two multiplexed libraries were sequenced in a single lane using Hiseq2000 (Illumina) with 101 bp paired-end reads. The short reads were modified by removing adapter sequences and were trimmed to remove low-quality sequences shorter than 64 bp and below QV (quality value) 20. The cleaned high-quality reads were mapped on the Tasmanian devil nuclear genome (Devil_ref v7.0: GCA_000189315.1) and mitochondrion genome (FN666604) using bowtie2 software [13], with three mismatches being allowed in the software. The quality of mapping sequences was evaluated by SAMtools [14], and the mapped sequences were visualized using Integrative Genomics Viewer [15]. The short-read sequences were also de novo-assembled using Trinity [16].

Transcripts were assembled and annotated using Cufflinks [17] and quantified with RSEM software [18]. Fragments per kb of exon per million mapped fragments (FPKM) and reads per kb of exon per million mapped reads (RPKM) were calculated, and genes expressed differentially between samples were found using Cuffdiff [17].

Appendix A shows the number of RNA-seq reads in the Tasmanian devil testis and brain. We identified 42,964 genes in total, of which 16,438 transcripts were previously reported by Murchison et al. (2010) [10]. Of the 42,964 genes, 20,414 loci are registered in Ensembl.

### 2.3. Comparative Evolutionary Analyses

We identified 1:1 orthologous genes among 10 species and created a testis expression gene list of 5627 orthologous genes expressed in the testis from at least one other of the 10 species. The orthologous list was created using Ensembl biomart. The 10 species are Tasmanian devil in this study and nine species from a previous study [9]: human (*Homo sapiens*), chimpanzee (*Pan troglodytes*), bonobo (*Pan paniscus*), gorilla (*Gorilla gorilla*), rhesus macaque (*Macaca mulatta*), mouse (*Mus musculus*), opossum (*Monodelphis domestica*), platypus (*Ornithorhynchus anatinus*), and chicken (*Gallus gallus*). If the RPKM value of a gene is <1, the gene is considered not to be expressed [19]. Thus, we assume that RPKM < 1 indicates loss of the gene’s expression, and RPKM ≥ 1 indicates acquisition of the gene’s expression. All RPKM values can be requestable. Of the 5627 genes, 2989 are expressed in the testes of all 10 species (RPKM ≥ 1).

Our comparison showed that the 22 genes acquired testis expression in the stem lineage of marsupials based on bulk RNA-seq data. In human testes, however, expression of one of the 22 marsupial testis-expressed genes, *PRDM1* (1.2 nTPM: normalized transcripts per million), is shown, and medium (25–75% fraction of stained cells) protein expression is reported as well (the Human Protein Atlas: https://www.proteinatlas.org/, accessed on 29 November 2023). Since we categorized the expressed genes into a binary state (present or non-present) based on an expression level (RPKM ≥ 1 or RPKM < 1), genes situated in the cut-off limit could be identified as non-expressed. In *PRDM1*, for example, its eutherian RPKM is 0.53 ± 0.34 (human; RPKM = 0.80 ± 0.50, TPM = 1.75 ± 0.37), suggesting it is not likely expressed in this category, although the expression level is slightly below the cut-off. Using this cut-off, we found that the expression level of eutherian *PRDM1* is much lower than the marsupial expression (RPKM = 192.02 ± 130.21), while *IGFBP1* is detected in *Sertori* cells by only human single-cell RNA sequencing since the single-cell RNA-seq is known to be highly sensitive, more than the bulk RNA-seq (the Human Protein Atlas).

### 2.4. Molecular Evolutionary Analyses

The translated nucleotide sequences were aligned using multiple sequence comparisons by log expectation (MUSCLE) [20], and the entire alignment is available in the Appendix A. Phylogenetic trees were constructed using neighbor-joining (NJ) and maximum likelihood (ML) methods implemented in the MEGA7 program [21]. The reliability of the trees was assessed by bootstrap re-sampling with 1000 replications.

## 3. Results and Discussion

### 3.1. The Evolutionary Tempo and Mode of Testis Genes in Theria

First, we identified 2989 testis-expressed genes that are evolutionary conserved in 10 species using RNA-seq data (see Section 2). Of 2989 testis-expressed genes in mammals as well as chickens, 53 are expressed in testes of the mammals (eutherians, marsupials, and monotremes) only (Section 2, Figure 1, and Appendix A). Of 53, 22 are expressed in adult testes of the marsupials, and this means that the testis expression of these genes was acquired in the stem lineage of marsupials (Figure 1 and Table 1). The 22 genes are located on the autosome and include a serine protease (*TRAF3IP3*), protease inhibitors (*IGFBP1*, *SERPINA10*), a glycosyltransferase (*FUT9*), a zinc finger transcription factor (*BCL11B)*, an oxidase (*TYRP1*), an actin-binding protein (*LMOD3*), a deaminase (*AMPD1*), and a signaling molecule (*DAB1*) (Table 1).

There are no genes expressed commonly in the testes of only the eutherians (Figure 1). One reason for this may be a rapid evolution of gene expression in eutherians. If gene expression has changed rapidly in one group, common gene expression in all groups cannot be found from the present transcriptomic information. To test whether the rapid evolution of genes has occurred in eutherians, we investigated phylogenetic trees of genes shared by Theria. Fifteen genes were expressed in the testes of only the therian species, and we inferred that this testis expression pattern had been obtained in the common ancestor of Theria (Table 2). We constructed phylogenetic trees of the 15 genes in 10 species used and calculated branch lengths in the eutherian and marsupial stem clades. In three genes (*ARHGAP28*, *SYNM*, and *PDZRN3*), the average of the eutherian branch lengths is significantly greater than that of the marsupial branch lengths (Figure 2A–C; Fisher’s exact test, *p* < 0.05). The longer branch lengths reflect faster gene evolution in Eutheria. In *Rho GTPase-activating protein 28* (*ARHGAP28*) the dn/ds ratio (0.39) at the eutherian stem lineage is significantly higher than 0.10 at the marsupial stem lineage (Figure 2A; Fisher’s exact test, *p* < 0.01). In the other two genes, the ratios are not different statistically (Figure 2B,C).

These observations are consistent with previous reports: in rodents and primates, the evolution of testis-expressed genes is accelerated compared to that of genes expressed in other tissues [22]. In our study, the rapid evolution of genes is not observed in marsupials. The evolutionary tempo and mode of testis-expressed genes may be different in marsupials and eutherians.

### 3.2. Therian-Specific Testis-Expressed Genes

Our results support the hypothesis that at least 15 genes obtained testis expression in the common ancestor of Theria, and then three (*ARHGAP28*, *SYNM*, and *PDZRN3*) evolved rapidly in only the eutherians (Table 2, Figure 1 and Figure 2). One reason is that the mutation rate at the genome level is lower in marsupials than in eutherians. Feigin et al. (2018) estimated that the mutation rate is 1.17 × 10^−9^ mutations per base per generation in the Tasmanian devil genome, compared with 6.98 × 10^−8^ mutations per base per generation in the human genome [23]. Our phylogenetic analyses suggest that the evolutionary rate of the three genes did not slow down in the stem lineage of marsupials compared to the ancestral state of Theria; rather, the evolutionary rate increased in the stem lineage of eutherians. However, in *ARHGAP28* the increasing rate (longer branch in eutherians) is more likely the acceleration of the nonsynonymous substitution rate due to adaptive evolution in the stem lineage of eutherians. The phylogenetic tree using the amino acid sequences shows that the branch at the eutherian stem is six times longer than that at the marsupial stem (Appendix A; Fisher’s exact test, *p* < 0.01), and this is consistent with the higher dn/ds ratio at the branch of the eutherian stem (Figure 2A). *ARHGAP28* is one of the GTPase-activating proteins (GAPs) that activate small monomeric Rho GTPases, and it is highly expressed in developing eutherian testes [24,25,26]. In the eutherian ancestor, ARHGAP28 may have acquired adaptive changes that enabled it to control molecules expressed in testis but the precise function of ARHGAP28 in testis has not yet been studied.

*Synemin* (*SYNM*) is a member of the intermediate filament family and encodes a 230-kDa polypeptide having a cytoskeletal role mainly in muscle [27]. *PDZ domain containing ring finger 3* (*PDZRN3*) is a ubiquitin ligase required for vascular morphogenesis in endothelium [28]. These ubiquitous genes are likely expressed not only in testis (mRNA/protein data available from the Human Protein Atlas database, accessed on 27 October 2022), although they probably have a fundamental function in testes. *Cytochrome P450 family 2 subfamily R member 1* (*CYP2R1*) is also a ubiquitous gene listed in the therian testis-expressed genes (Table 2). CYP2R1, also known as vitamin D 25-hydroxylase, acts in the vitamin D synthesis pathway and drug metabolism, and vitamin D is positively associated with sperm motility, affecting male fertility in humans [29].

### 3.3. 22 Genes: Functional Differentiation in a Member of a Large Gene Family

Of the 22 marsupial and 15 therian testis-expressed genes, four are type II cadherin genes (*CDH7*, *18*, *19*, and *20*). *CDH9* is also one of the platypus testis-expressed genes (Appendix A). Cadherin is a transmembrane or membrane-associated glycoprotein that mediates cell adhesion in a Ca^2+^-dependent manner and is essential for spermatogenesis [30]. Type II cadherin proteins do not contain a conserved cell adhesion recognition sequence, His-Ala-Val, in their ectodomain module [31]. Multiple members of the cadherin family (*CDH6, 8, 10*, and *11*) are expressed in the rat testis [32] and are involved in germ cell adhesion to *Sertoli* cells [33,34]. However, the function of type II cadherin in testes is not known.

As noted above, protease inhibitors (*IGFBP1*, *SERPINA10*) and a serine protease (*TRAF3IP3*) are included in the marsupial testis-expressed genes and may be required for spermatogenesis [35]. *SERPINA* comprises a large gene family, one of whose members *SERPINA5* (also known as protein C inhibitor) is known to be upregulated by testosterone [35]. Testis expression of the gene is shared only by therian mammals, as the chicken *SERPINA5* is expressed in the liver and kidney. While the marsupial *SERPINA10* is expressed in testes, the human and chicken genes are highly expressed in the liver (AceView) [36]. This implies that in the therian ancestor, *SERPINA5* acquired a new function as a testis-expressed gene, and *SERPINA10* became a testis-specific protease in the marsupial ancestor. A 5,6-dihydroxyindole-2-carboxylic acid oxidase (*TYRP1*) and an actin-binding protein (*LMOD3*) are the two other marsupial testis-expressed genes. It has been suggested that male infertility is caused by oxidative stress and downregulation or mutations of actin-related genes (e.g., *ACTL7B*) in sperms [37,38,39].

These 22 genes newly acquired testis expression in the stem lineage of marsupials, but their function in testis is not yet known. Each of the genes is a member of a gene family, one of whose other members reportedly have functions in testis differentiation and spermatogenesis of eutherians. The particular subsets of gene family members that function in testis may differ slightly between marsupials and eutherians, and these functional differences between genes may contribute to the different patterns of sex differentiation observed in marsupials and eutherians.

### 3.4. Perspectives on the Function of PRDM1

Using previous data [40], we investigated whether the 22 marsupial-adult testis-expressed genes are expressed in opossum newborn testes, and one gene, *PR/SET Domain 1* (*PRDM1*), was also found to be expressed in the neonatal period (postnatal 6 days; RPKM = 18.59 ± 10.41). *PRDM* is a gene family with an ancient origin, and most of its members emerged early in animal evolution [41,42]. The number of PRDM genes is 13–15 in marsupials and 12–17 in eutherians [41]. The paralogues to *PRDM1* are crucial for the development of testis. *PRDM14* is expressed in mouse PGCs, and its knockout mice are infertile [43]. *PRDM8* regulates mouse testis steroidogenesis by repressing *p450c17* (*CYP17A1*) and *luteinizing hormone receptor* (*LHR*) [44]. The accelerated evolution of *PRDM9* is supported by site-specific positive selection [45], but we could not identify any statistically significant positive selection sites in *PRDM1.*

PRDM1 is a transcription factor with five zinc finger motifs and represses *β-interferon* (*β-INF*) gene expression by binding to the positive regulatory domain 1 (PRD1) element of the *β-INF* promoter [46]. In humans and mice, *PRDM1* is highly expressed in immune system cells such as B/T lymphocytes and natural killer (NK) cells. In mouse studies, *β-INF* overexpression in the testis disrupts spermatogenesis [47]. PRDM1 is also known as B-lymphocyte-induced maturation protein 1 (Bimp1) and is crucial for primordial germ cell (PGC) differentiation [48]. The function of *PRDM1* in testes is not well understood, and their functional analysis is necessary in the future.

## 4. Conclusions

We show the evolutionary tempo of three genes (*ARHGAP28*, *SYNM*, and *PDZRN3*) is fast in eutherians compared to marsupials, and especially the amino acid substitutions of *ARHGAP28* are accumulated in the common ancestor of eutherians. Those three genes are probably involved in the specificity of eutherian testes. We also identified 22 genes that are involved in the specificity of marsupial adult testes with high expression based on the comparative bulk RNA-seq data. We observed the functional differentiation and sub-functionalization, that is, marsupial-specific high testis expression, of a member of a large gene family such as *PRDM*, *CDH*, and *SERPINA*.

## Figures and Tables

**Figure 1 animals-14-02316-f001:**
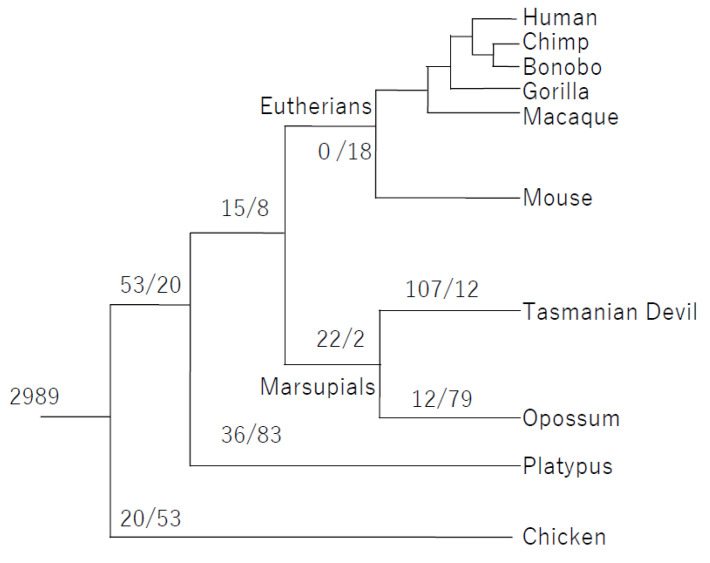
Acquisition/loss of testis-expressed genes in 10 species. The genes at each clade are listed in Table 1, Table 2, Appendix A, and the left and right of the ‘/’ symbols mean acquisition and loss of testis-expressed genes, respectively.

**Figure 2 animals-14-02316-f002:**
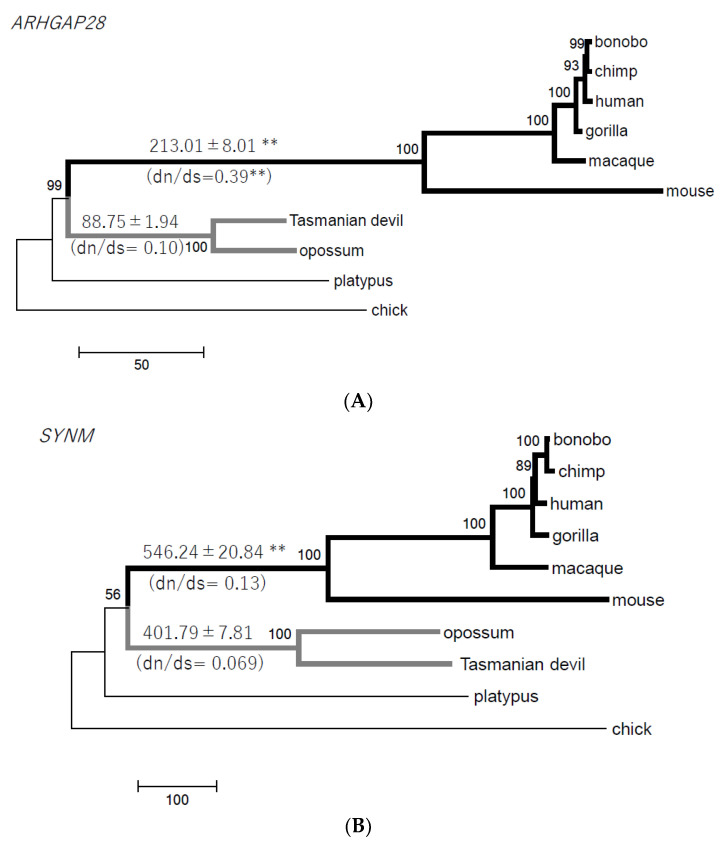
The phylogenetic trees, branch lengths, and dn/ds ratios for three therian-specific testis genes in 10 species. (**A**) *ARHGAP28* genes (1833 bp). (**B**) *SYNM* genes (2235 bp). (**C**) *PDZRN3* genes (2256 bp). The trees were constructed by the NJ method using the number of nucleotide differences. The average and standard deviation of branch lengths in six eutherians are shown above the black bold branch, and those in two marsupials are shown above the gray bold branch. The dn/ds ratio in six eutherian or two marsupial pairs is shown under the branch. ** and * mean *p* < 0.01 and *p* < 0.05, respectively, and are supported by Fisher’s exact test.

**Table 1 animals-14-02316-t001:** Marsupial-specific testis-expressed genes.

NIPAL1	NIPA-like domain containing 1
LMOD3	Leiomodin 3
FUT9	Fucosyltransferase 9
TECTB	Tectorin beta
AMPD1	Adenosine monophosphate deaminase 1
TYRP1	Tyrosinase-related protein 1
PRDM1	PR/SET domain 1
BCL11B	B-cell CLL/lymphoma 11B
BEST3	Bestrophin 3
IHH	Indian hedgehog
IKBKE	Inhibitor of nuclear factor kappa B kinase subunit epsilon
DAB1	DAB1, reelin adaptor protein
TRAF3IP	TRAF3-interacting protein 3
RBM20	RNA-binding motif protein 20
SERPINA10	Serpin family A member 10
CDH7	Cadherin 7
CDH19	Cadherin 19
CDH20	Cadherin 20
IGFBP1	Insulin-like growth factor binding protein 1
ZNF750	Zinc finger protein 750
THEMIS	Thymocyte selection associated
NTS	Neurotensin

**Table 2 animals-14-02316-t002:** Therian-specific testis-expressed genes.

ARHGAP28	Rho GTPase-activating protein 28
CYP2R1	Cytochrome P450 family 2 subfamily R member 1
ESRP2	Epithelial splicing regulatory protein 2
ATP9A	ATPase phospholipid transporting 9A
MYO15A	Myosin XVA
HSPA12A	Heat shock protein family A (Hsp70) member 12A
TOGARAM2	TOG array regulator of axonemal microtubules 2
PDZRN3	PDZ domain containing ring finger 3
SLC7A2	Solute carrier family 7 member 2
SYNM	Synemin
CDH18	Cadherin 18
GJA5	Gap junction protein, alpha 5
SH3RF2	SH3 domain containing ring finger 2
WFDC1	WAP four-disulfide core domain 1
ELFN2	Extracellular leucine rich repeat and fibronectin type III domain containing 2

## Data Availability

The data used were deposited in a repository (DRA016090).

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
