# Peer review of "Adaptive Evolution and Functional Differentiation of Testis-Expressed Genes in Theria"

_animals, 2024, doi:10.3390/ani14162316_

Round 1

Reviewer 1 Report

Comments and Suggestions for Authors

Simple summary: well written, won't suggest anything here.

Abstract: Precisely summarized.

Introduction: No comments

Material and methods: In line 81 authors used brain tissue for their experiment. Was brain used as control? I did not see mentioning about it.

In line 113 authors mentioned RPKM values and how they used to to determine expressions. This will help the readers to understand the results. Good work.

Results and discussions: Did you use platypus genome too? It is in results but not in material and methods.

Are there any gene expression you found evolved differently in marsupials? I saw genes differently expressed in eutherians, but not in marsupials. Since that was the goal of this study that's why asked. 

Conclusion: no comments.

Author Response

Comments of Reviewer 1:

Material and methods: In line 81 authors used brain tissue for their experiment. Was brain used as control? I did not see mentioning about it.

>Thank you very much for your comments. We used brain sample for RNA-seq, but do not use it in the result. I removed that.

Results and discussions: Did you use platypus genome too? It is in results but not in material and methods.

>We used platypus genome, and it is written in line 115.

Are there any gene expression you found evolved differently in marsupials? I saw genes differently expressed in eutherians, but not in marsupials. Since that was the goal of this study that's why asked. 

>In this analysis, we did not find adaptive evolution of genes in a marsupial-specific way, but we found 22 genes which are highly expressed in only marsupials. It is written in line 146-152 and table 1. We discuss the 22 genes in 3.3 and 3.4 (Results and Discussion).

Reviewer 2 Report

Comments and Suggestions for Authors

Review on the manuscript titled “Adaptive evolution and functional differentiation of testis-expressed genes in Theria” by Katsura et al., 2024.

                The authors performed evolutionary analysis of testis-specific genes based on RNA-seq analysis and outlined the events taken place in the course of evolution. In particular, they correspond 22 marsupial and 15 therian –specific genes based on gene expression rate.

                The authors provided explicit ML trees on the Gain-Loss genes evolutionary structure in the course of evolution (Figs 1, 2). They as well they correspond the reported high evolutionary rates of testis-specific genes as reported before (Khaitovich et al., 2006); chapter 3.1. They found the mammals acquired 53 genes, 22 of them are marsupial specific, compared to outgroup species (Fig. 1, Chicken).

                They reported that “evolutionary tempo of three genes (ARHGAP28, SYNM, and PDZRN3) is fast in eutherians compared to marsupials”. As a conclusion, they state that “We observed the functional differentiation and sub-functionalization, that is marsupial-specific high testis expression, of a member of a large gene family such as PRDM, CDH, and SERPINA”.

                Overall, the manuscript is well structured and transparent. Inferences on the testis genes evolution dynamics, including genes expression rate and content elucidated by the authors is quite relevant ones and confident enough. The study would be definitely of interest to the researchers in the field. Some notes are presented below.

Notes.

1)      The authors skipped the paper of Murat et al. (Nature volume 613pages 308–316 (2023)), where marsupials have also been considered. It’d be relevant to mention/discuss it in the manuscript.

2)      The supplementary data should be reformed. Table S1 is not immediately comprehensive as is since there is no column labels. Also, why the authors didn’t provide the actual number of genes with >1, <1 RPKM, and instead, using symbols instead? Tables S2-S4 would be more relevant with added rpkm values and deposited in excel spreadsheets. Alignment files should be declared in the text as Information sheets named in the supplementary file. I didn’t find Video S1: title, probably it’s redundant?

3)      The  genes sets outlined are rather sparse of interconnection rate based on string-db.org suite analysis implying many different pathways involved. There is only cadherin/catenin genes triad network that is definitely outlined in marsupials by string-db, and none in eutherians. While the authors explicitly describe genes in the final sections (3.2-3.4), it’d be good to assess the networks with relevant background genes encompassing the set. For example, PRDM1 gene in marsupials (opossum) is associated with chromatin rearrangement genes based on string-db data in opossum that is not outlined in chapter 3.4 (see the attached word file).

I suggest the key genes (PRDM1, ARHGAP28, etc) should be presented (and desirably supported by RNA-seq data of corresponding genes) with closest neighboring ones using string-db suite or other one. Genes network covariance would also yield higher confidence in target gene expression alteration. While this section is not rather evolution oriented, it might be posted to the supplementary.

Author Response

Comments of Reviewer 2:

  • The authors skipped the paper of Murat et al. (Naturevolume 613, pages 308–316 (2023)), where marsupials have also been considered. It’d be relevant to mention/discuss it in the manuscript.

> Thank you very much for your comments. The paper of Murat et al. have done single cell RNA-seq, and we used bulk RNA-seq. Those data cannot be directly compared, but we mentioned their paper in introduction in line 69-71. Murat et al. focused on primate or placental mammalian evolution of testes expressed genes and used single marsupial species and cannot mention the evolution of marsupials.

  • The supplementary data should be reformed. Table S1 is not immediately comprehensive as is since there is no column labels. Also, why the authors didn’t provide the actual number of genes with >1, <1 RPKM, and instead, using symbols instead? Tables S2-S4 would be more relevant with added rpkm values and deposited in excel spreadsheets. Alignment files should be declared in the text as Information sheets named in the supplementary file.

>Tables S1 was changed as the reviewer 2 mentioned. However, the actual number of genes is shown in Figure 1, and we did not change O or X. We deposited an excel file, but it was exchanged to a PDF file because of the journal rules. The rpkm values can be requestable. In line 134, the name of supplementary file was also changed.

I didn’t find Video S1: title, probably it’s redundant?

> We did not understand what “Video S1” is, and delated it.

  • The  genes sets outlined are rather sparse of interconnection rate based on string-db.org suite analysis implying many different pathways involved. There is only cadherin/catenin genes triad network that is definitely outlined in marsupials by string-db, and none in eutherians. While the authors explicitly describe genes in the final sections (3.2-3.4), it’d be good to assess the networks with relevant background genes encompassing the set. For example, PRDM1 gene in marsupials (opossum) is associated with chromatin rearrangement genes based on string-db data in opossum that is not outlined in chapter 3.4 (see the attached word file).

>Thank you very much for the suggestion. We checked the website of “string-db”, but we could not find any network of PRDM1 gene in testes and could not find any interaction among the genes mentioned in the manuscript. The network is interesting as reviewer 2 mentioned, but we will not mention string-db data since this type of data also includes a false discovery.